# An Expressive and Self-Adaptive Dynamical System for Efficient Function Learning

**Chuan Liu** [1]  **Chunshu Wu** [1]  **Ruibing Song** [1]  **Ang Li** [2 3]  **Ying Nian Wu** [4]  **Tony (Tong) Geng** [1 5]

## Abstract

Function learning forms the foundation of numerous scientific and engineering tasks. While modern machine learning (ML) methods model complex functions effectively, their escalating complexity and computational demands pose challenges to efficient deployment. In contrast, natural dynamical systems exhibit remarkable computational efficiency in representing and solving complex functions. However, existing dynamical system approaches are limited by low expressivity and inefficient training. To this end, we propose EADS, an Expressive and self-Adaptive Dynamical System capable of accurately learning a wide spectrum of functions with extraordinary efficiency. Specifically, (1) drawing inspiration from biological dynamical systems, we integrate hierarchical architectures and heterogeneous dynamics into EADS, significantly enhancing its capacity to represent complex functions. (2) We propose an on-device training method that leverages intrinsic electrical signals to update parameters, making EADS self-adaptive with exceptional efficiency. Experimental results across diverse domains demonstrate that EADS achieves higher accuracy than existing works, while offering orders-of-magnitude speedups over traditional neural network solutions on GPUs for both inference and training, showcasing its broader impact in overcoming computational bottlenecks across various fields.

## 1. Introduction

Function learning is essential for modeling, analysis, and prediction across a wide range of scientific and engineering tasks. Modern ML methods, particularly neural networks (NNs), have demonstrated exceptional capability in approximating complex functions by learning from data. Despite their remarkable achievements, the computational demands of ML models have soared due to the escalating model complexity. Even on the most powerful GPU, training these models remains prohibitively expensive, and the waning of Moore's Law exacerbates this challenge. As a result, the quest for alternative, efficient computational paradigms has become increasingly urgent.

Nature offers an elegant remedy to the growing computational burden of modern ML methods. Natural dynamical systems exemplify how complex functions can be efficiently modeled and solved through their intrinsic processes. For example, partial differential equations (PDEs) that govern physical phenomena or chemical reactions are inherently solved by the dynamical systems they describe. This natural process can be interpreted as the system evolving over an energy landscape, where lower energy states correspond to higher statistical likelihood. Driven by thermodynamic principles, such systems instinctively evolve toward the equilibrium state, generating optimal solutions. This process, which we refer to as ***natural annealing***, exhibits exceptional efficiency. Motivated by this potential, this work investigates whether dynamical systems can be leveraged to develop a novel ML paradigm that effectively learns various functions with significantly improved efficiency.

Although early theoretical work highlighted the promise of dynamical systems for ML (Weinan, 2017; Li & Weinan, 2021; Weinan et al., 2022), practical progress has been hindered by the lack of suitable hardware embodiments. Fortunately, recent breakthroughs in programmable electronic dynamical systems (Afoakwa et al., 2021; Böhm et al., 2022) have marked a turning point. Leveraging these electronic dynamical systems, previous studies have demonstrated that the power of nature can be leveraged to tackle some simple learning problems with remarkable efficiency (Wu et al., 2024; Song et al., 2024b; Böhm et al., 2022). However, the applicability and broader impact of electronic dynamical

---

[1]University of Rochester, Rochester, NY, USA [2]Pacific Northwest National Laboratory, Richland, WA, USA [3]University of Washington, Seattle, WA, USA [4]University of California Los Angeles, Los Angeles, CA, USA [5]Rice University, Houston, TX, USA. Correspondence to: Tony (Tong) Geng <tg62@rice.edu>.

*Proceedings of the 42nd International Conference on Machine Learning*, Vancouver, Canada. PMLR 267, 2025. Copyright 2025 by the author(s).

systems remain significantly limited due to two key challenges: *1. Low Expressivity:* Existing dynamical systems are governed by energy functions that entail simple interactions among nodes, and hence limiting their capacity to represent complex functions. *2. Inefficient Training:* Existing approaches realize inference on dynamical systems through on-device natural annealing; however, the training process to construct the desired energy landscape is still performed on digital processors, resulting in high training costs. This decoupling of training and inference deviates from the intelligence observed in natural systems, preventing this emerging ML paradigm from addressing the most pressing computational challenge in ML development: extremely high training costs. Therefore, substantial advancements are needed to fully realize the potential of dynamical systems.

Notably, numerous scientific studies (Wills et al., 2005; Friston, 2010; Inagaki et al., 2019) suggest that the brain also functions as a dynamical system, performing inference and training in a collocated manner. Inspired by the brain, a highly efficient and powerful dynamical system, we propose enhancing the electronic dynamical system in two ways: (1) improving the system's expressivity through hierarchical architectures and heterogeneous dynamics; (2) enabling on-device training to fully leverage its extraordinary computational power. Specifically, to enhance expressivity, we improve the dynamical system with a hierarchical structure and heterogeneous dynamics, facilitating progressive information refinement through distinct processing stages. Furthermore, we propose a learning method that allows the dynamical system to leverage its intrinsic electrical signals to self-construct its energy landscape, align with target data distributions, and achieve on-device training with exceptional efficiency.

To this end, this work introduces EADS, a nature-powered ML paradigm that leverages the computational power of dynamical systems for accurate and efficient function learning. By expanding the applicability of dynamical systems to encompass functions from diverse domains, e.g. real-world problems, PDEs in scientific computing, and ML kernels, EADS holds the potential to overcome persistent computational bottlenecks and drive advancements across various fields. The overview of EADS is shown in Figure 1, and the contributions of this paper are summarized as:

- We propose EADS, an expressive and self-adaptive dynamical system capable of accurately and efficiently learning functions across diverse domains.

- We introduce hierarchical structures and heterogeneous dynamics to enhance the dynamical system's capacity to represent complex functions.

- We propose an on-device training method that enables the dynamical system to train its parameters using in-

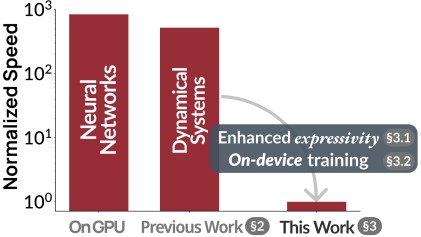

*Figure 1.* The overview of EADS.

ternal electrical signals with exceptional efficiency.

- Experimental results demonstrate that EADS accurately learns various functions, achieving orders-of-magnitude speedups ($10^3 \times$) over baselines on GPUs.

## 2. Preliminaries and Related Work

This section provides preliminaries of ML through the lens of dynamical systems, including both theoretical developments and recent advances with hardware embodiments. Subsequently, we review related work that uses dynamical systems to address various tasks, from combinatorial optimization to advanced ML applications.

### 2.1. Preliminaries

**ML via Dynamical Systems.** The potential of dynamical systems in ML was highlighted by (Weinan, 2017; Li & Weinan, 2021; Weinan et al., 2022), demonstrating their ability to model complex, high-dimensional nonlinear functions through continuous transformations. These works highlight that deep NNs succeed by composing simple functions to approximate complex ones, while dynamical systems extend this compositional approach to an infinitesimal limit. Compared to deep NNs, dynamical systems offer several advantages: (1) greater flexibility in imposing constraints and incorporating domain-specific structures, facilitating more transparent theoretical analysis than purely discrete-layer architectures; and (2) easier integration of ML techniques with physical models, enabling seamless interaction with real-world physical processes. Despite these promising theoretical advancements, practical adoption of dynamical systems in ML has been limited by the lack of suitable hardware embodiments.

Fortunately, recent advancements in programmable electronic dynamical systems have revived interest in this field. Originally conceived as physical embodiments of the binary Ising model for solving binary combinatorial optimization problems, these systems have since expanded to tackle binary learning tasks (Pan et al., 2023; Liu et al., 2023). (Wu et al., 2024) later extended the binary Ising model to support real-valued nodes, enabling real-valued graph learning tasks

(Liu et al., 2025b). This extension results in the following energy function (also referred to as Hamiltonian):

$$\mathcal{H}_{rv} = -\sum_{i \neq j}^{N} J_{ij} x_i x_j + \sum_{i=1}^{N} h_i x_i^2, \quad x_i, x_j \in \mathbb{R}. \quad (1)$$

Here, $J_{ij}$ represents the interaction strength between nodes $x_i$ and $x_j$, while $h_i$ denotes the self-reaction strength of $x_i$ to external influences. Assuming a Boltzmann distribution $p_{rv} = e^{-\beta \mathcal{H}_{rv}}/Z$, where the partition function $Z$ serves as a normalization constant that ensures that the probabilities sum up to one, the energy landscape is mapped to a probability distribution, with the lowest energy state corresponding to the highest probability state. The system's dynamics are designed as:

$$\frac{dx_i}{dt} = -\frac{\partial \mathcal{H}_{rv}}{\partial x_i} = \sum_{j \neq i}^{N} \left( J_{ij} + J_{ji} \right) x_j - 2 h_i x_i, \quad (2)$$

which guarantees the spontaneous energy decrease of the system:

$$\frac{d\mathcal{H}_{rv}}{dt} = \sum_{i=1}^{N} \left( \frac{\partial \mathcal{H}_{rv}}{\partial x_i} \frac{dx_i}{dt} \right) = -\sum_{i=1}^{N} \left( \frac{\partial \mathcal{H}_{rv}}{\partial x_i} \right)^2 \leq 0. \quad (3)$$

When applied to graph learning tasks, a subset of nodes is fixed to input values, while the remaining nodes, serving as output nodes, are randomly initialized and evolve according to the designed dynamics. Given a well-trained Hamiltonian that accurately captures the correlation between inputs and outputs, the spontaneous energy decrease makes the system instantly anneal to desired solutions.

**Physical Embodiment of Dynamical Systems.** This dynamical system is physically realized using programmable electronic components, such as resistors and capacitors, as illustrated in Figure 2. The key idea behind this embodiment is to precisely and efficiently realize the node dynamics using electronic components. In this design, each node $x_i$ is implemented as a nanoscale capacitor within a node unit ($N_i$), with its voltage representing the node value. Each capacitor is coupled with a resistor of resistance $R_i = 1/(2h_i)$, forming a resistive current within the node unit, realizing the term $2h_i x_i$ in the node dynamics. Additionally, capacitors from different node units ($N_i$ and $N_j$) are structurally connected via a programmable resistor in the coupling unit ($CU_{ij}$), with resistance $R_{ij} = 1/J_{ij}$. This configuration effectively incorporates the term $\sum_{j \neq i}^{N} \left( J_{ij} + J_{ji} \right) x_j$ in the node dynamics, implementing a resistively coupled capacitor network.

**Offline Training of Dynamical Systems.** Training a dynamical system involves optimizing the parameters $\mathbf{J}$ and $\mathbf{h}$ in the Hamiltonian $\mathcal{H}_{rv}$ to construct an energy landscape that reflects the target data distribution. Previous works have

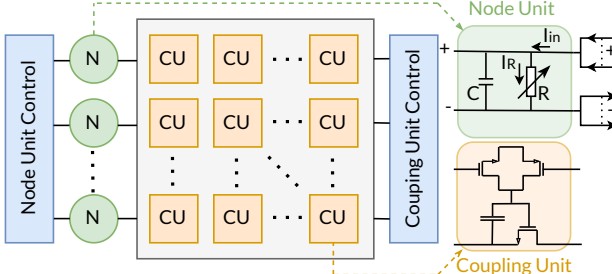

*Figure 2.* The backbone electronic dynamical system.

trained the model using computationally expensive statistical methods executed on digital processors, mainly GPUs. Specifically, the training process begins by estimating the node values using methods such as conditional likelihood maximization (Wu et al., 2024). The discrepancies between the estimations and the ground truths are evaluated using metrics such as Mean Absolute Error (MAE). These metrics serve as loss functions to update the model parameters, thereby reconstructing the energy landscape to align with the data distribution. During inference, the well-trained parameters are mapped onto the circuit, then natural annealing drives the system toward the lowest energy state, enabling it to find the solution with the highest probability for the target problem.

## 2.2. Related Work

Dynamical systems have gained significant attention as an efficient computing paradigm, initially applied to optimization problems (Sharma et al., 2023a; Sun et al., 2025) and subsequently extended to machine learning tasks (Wu et al., 2024; Liu et al., 2025a; Song et al., 2024a). The Ising machine, one of the earliest hardware implementations leveraging dynamical systems, embodies the Ising model originally developed for ferromagnetism in statistical physics. Ising machines have demonstrated breakthrough efficiency in solving numerous binary optimization problems (Böhm et al., 2019; Mohseni et al., 2022; Lo et al., 2023). For instance, researchers have employed Ising machines to address satisfiability (SAT) problems (Sharma et al., 2023a;b; Sun et al., 2025), as well as MAX-CUT and graph coloring problems (Wang & Roychowdhury, 2019; Böhm et al., 2019; Liu et al., 2025c).

Recognizing their potential, researchers have explored dynamical systems in ML applications such as unsupervised NN training (Böhm et al., 2022), graph learning (Pan et al., 2023), and collaborative filtering (Liu et al., 2023). While these studies provide valuable insights into leveraging dynamical systems for ML tasks, their scope and applicability are limited by the binary nodes of Ising machines, hindering progress in more complex, real-valued scenarios. Recent work (Wu et al., 2024; Wu et al.) introduced real-valued

dynamical systems to accelerate inference in graph learning problems; however, their contributions are constrained by two key limitations. First, while the proposed Hamiltonian supports real-valued nodes, it only accounts for simple interactions, limiting its ability to capture the intricate patterns present in many complex problems. Second, their approach utilizes the power of dynamical systems exclusively during the inference phase, leaving the computationally intensive training process unaddressed. Although some simple on-device training methods have been developed (Liu et al., 2025a; Wu et al., 2025), they have only demonstrated effectiveness on simple models. These limitations, which significantly constrain the broader impact of dynamical systems, will be addressed in this work.

## 3. Methodology

The pursuit of powerful and highly efficient computing systems has been profoundly influenced by the extraordinary capabilities of biological systems, particularly the human brain. By contrasting the brain's remarkable capabilities with existing physically embodied dynamical systems, two fundamental limitations emerge: (1) insufficient expressivity and (2) inefficient training. This section introduces solutions to address these limitations. In particular, Section 3.1 introduces a hierarchical, heterogeneous dynamical system to boost expressivity, and Section 3.2 presents an on-device training method that makes the system self-adaptive with exceptional efficiency.

### 3.1. Expressivity Enhancement

Existing dynamical systems with physical embodiments, while promising, exhibit several limitations when compared to the brain. (1) Flat Structure. Unlike the brain's hierarchical organization, which processes information through multiple layers of increasing abstraction, existing dynamical systems maintain a flat structure. (2) Homogeneous Dynamics. In contrast to the brain's rich repertoire of nonlinear processing mechanisms, where different regions exhibit diverse dynamics, current dynamical systems rely on uniform dynamics, characterized by linear interactions among nodes. These constraints limit their ability to model intricate, nonlinear functions.

**Brain-Inspired Enhancements.** To address these limitations, we propose two key enhancements inspired by the architecture and functionality of the brain: (1) a hierarchical structure and (2) heterogeneous dynamics. Our enhanced system implements a multi-stage processing pipeline inspired by the brain's information processing mechanisms.

The enhanced system initiates with a projection that transforms inputs into an abstract hidden space, modeled by the following dynamics:

$$\frac{dh_i}{dt} = \sum_{j=1}^{N} P_{ij} x_j - r_i h_i, \tag{4}$$

where $x_j \in \mathbb{R}\,(j = 1, 2, \ldots, N)$ denotes the $j$-th input node, $h_i \in \mathbb{R}\,(i = 1, 2, \ldots, H)$ denotes the $i$-th hidden node. The projection weight $P_{ij} \in \mathbb{R}$ denotes the connection from $x_j$ to $h_i$, analogous to dendritic integration in biological neurons. The term $r_i$ denotes the self-reaction strength of the hidden node $h_i$.

Once the hidden nodes in Eq. 4 have stabilized, they further evolve through internal coupling, reflecting the dense local connectivity observed in cortical circuits:

$$\frac{dh_i}{dt} = \sigma \left( \sum_{k=1}^{H} J_{ik} h_k \right), \tag{5}$$

where $J_{ik} \in \mathbb{R}\,(i, k = 1, 2, \ldots, H)$ represents inter-node interaction weights, and $\sigma$ is a hardware friendly nonlinear function, such as ReLU. This nonlinear function can be efficiently implemented using diodes to regulate current flow, thereby enabling nonlinear processing capabilities.

Finally, the processed hidden states are mapped to the output space through:

$$\frac{dy_m}{dt} = \sum_{i=1}^{H} Q_{mi} h_i - r_m y_m, \tag{6}$$

where $y_m \in \mathbb{R}\,(m = 1, 2, \ldots, M)$ represents output nodes, $Q_{mi} \in \mathbb{R}$ is the output projection weight matrix, and $r_m$ denotes output self-reaction strength. This hierarchical pipeline yields significantly enriched expressivity than traditional flat and homogeneous dynamical systems, as demonstrated in Section 4.

**Physical Embodiment of Enhanced Dynamical Systems.** The physical implementation of the enhanced dynamical system builds upon the foundational design shown in Figure 2. Following the same design strategy, node values are mapped to capacitor voltages, enabling the natural realization of continuous-time dynamics. Each capacitor is coupled with a resistor of resistance $r_i$ or $r_m$, forming the resistive current within the node unit. The parameters $\mathbf{P}, \mathbf{J}, \mathbf{Q}$ are configured as conductances of programmable resistors, thereby implementing the system dynamics as electrical currents. Specifically, the terms $\sum_{j=1}^{N} P_{ij} x_j$, $\sigma\left(\sum_{k\neq i}^{H} J_{ik} h_k\right)$, and $\sum_{i=1}^{H} Q_{mi} h_i$ are mapped as the flow-in currents into the respective node units. The terms $r_i h_i$ and $r_m y_m$ are mapped as internal currents within each node unit. The design of input nodes $x_j$ and output nodes $y_m$ remains consistent with the original architecture, while the hidden nodes $h_i$ have been extended to incorporate the newly introduced dynamics, as illustrated in the "Improved design of $h_i$" section

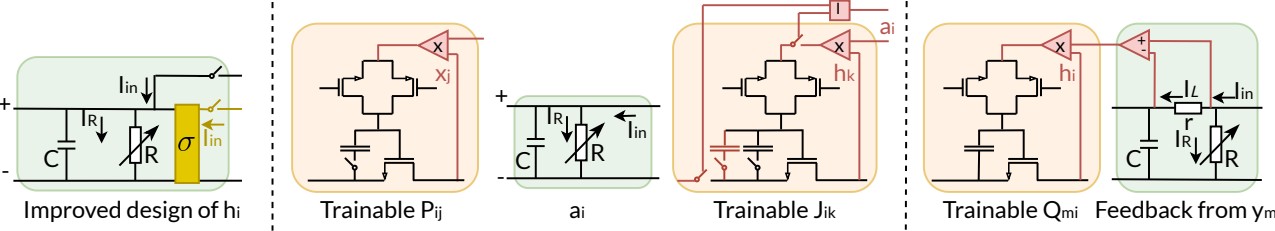

*Figure 3.* The key components of the enhanced dynamical system with on-device training support.

of Figure 3, where the added components are highlighted in yellow. As shown, two switches control the dynamics of each hidden node: when the black switch is closed, the circuit realizes the input-to-hidden dynamics described by Eq. 4; when the yellow switch is closed, it implements the inter-hidden dynamics of Eq. 5. The nonlinear function is implemented by using diodes to restrict the current, enabling effective hardware implementation of the nonlinearity.

### 3.2. Instant On-Device Training

Despite these enhancements, the system's advantages remain constrained without efficient training support, as training is the most computationally intensive process. Therefore, to extend the extraordinary computational power of dynamical systems to the training process, we propose an efficient on-device training method, *EC-Train*. This novel approach utilizes the intrinsic electrical signals of the dynamical system as feedback for on-device parameter adjustment, enabling self-adaptation to the target data distribution. EC-Train significantly reduces training costs, achieving orders-of-magnitude improvements in efficiency over conventional offline training on digital processors.

**On-Device Instant Training Approach.** According to fundamental physical principles (Kirchhoff's current law), each output node $y_m$ stabilizes when its flow-in current $I_m^{in} = \sum_{i=1}^{H} Q_{mi} h_i$ balances its internal resistor current $I_m^R = r_m y_m$. When $y_m$ is clamped to its ground truth value, the difference between $I_m^{in}$ and $I_m^R$ provides a direct measure of error. Consequently, the on-device training process of EC-Train aims to minimize the difference between $I_m^{in} - I_m^R$ for all output nodes when their values (voltages) set to ground truth values. In this way, the loss function of EC-Train can be formulated as:

$$L = \frac{1}{M} \sum_{m=1}^{M} (I_m^{in} - I_m^R)^2. \tag{7}$$

The error signal $\delta_m$ emerges intrinsically from the dynamical system, serving as natural feedback signals for parameter optimization:

$$\delta_m = \frac{2}{M} (I_m^{in} - I_m^R). \tag{8}$$

Specifically, the gradients with respect to $Q_{mi}$ are then

computed as: $\partial L / \partial Q_{mi} = \delta_m \cdot h_i$. Since output nodes $y_m$ are clamped to their ground truth values during training, $r_m$ essentially acts as a scaling factor for the ground truth signal. To achieve an efficient hardware implementation, we make $r_m = 1$, thereby streamlining both the training and inference processes.

However, for the inter-hidden-node coupling weights $J_{ik}$, we face a unique challenge: unlike output nodes, we lack ground truth values for hidden states $h_i$. To address this, we employ the Adjoint Sensitivity Method (Pontryagin, 2018), a powerful technique from optimal control theory that enables gradient computation through an auxiliary dynamical system. This approach is particularly suitable as it: (1) eliminates the need for ground truth hidden states, (2) maintains mathematical rigor while being hardware-realizable. Formally, we introduce an adjoint node $a_i = \partial L / \partial h_i$ for each hidden node $h_i$, with initial value being $\sum_m Q_{mi} \delta_m$ and dynamics governed by:

$$\frac{da_i}{dt} = -\sum_{l=1}^{H} a_l J_{li} I_l. \tag{9}$$

Here, $I_l$ is a boolean indicator defined as:

$$I_l = \sigma' \left( \sum_{s=1}^{H} J_{ls} h_s > 0 \right) = \mathbb{I} \left( \sum_{s=1}^{H} J_{ls} h_s > 0 \right). \tag{10}$$

The gradient with respect to coupling weights $J_{ik}$ is then computed through a dynamical process:

$$\frac{\partial L}{\partial J_{ik}} = -\int_{T}^{0} a_i I_i h_k dt, \tag{11}$$

which involves the evolution of $a_k$ and $h_i$ from $T$ backward to 0 to accumulate the updates.

For the parameters $P_{ij}$ and $r_i$ involved in the input-to-hidden projection stage, the stabilization of the hidden state $h_i$ follows Kirchhoff's current law. Specifically, $h_i$ reaches a steady state when its incoming current $I_i^{in} = \sum_{j=1}^{N} P_{ij} x_j$ is balanced by the current through its internal resistor, $I_i^R = r_i h_i$. Given that each input $x_j$ is clamped to its ground-truth value, the equilibrium condition implies a stable hidden state value of $h_i = \sum_{j=1}^{N} P_{ij} x_j$, assuming $r_i = 1$ without loss of generality, as it functions as a scaling factor. The error signal propagated from the inter-hidden

stage is denoted as $\delta_i = a_i|_{t=0}$. Consequently, the gradient with respect to the projection weight $P_{ij}$ is denoted as:

$$\partial L / \partial P_{ij} = \delta_i \cdot x_j. \tag{12}$$

Detailed derivations are provided in the Appendix.

**Physical Embodiment of EC-Train.** The proposed training approach introduces simple yet effective hardware modifications that enable self-training through electrical current feedback. As shown in Figure 3 (highlighted in red), we introduce additional feedback signal paths for each parameter. These feedback paths allow the electronic dynamical system to propagate signals to the coupling units, facilitating parameter adjustments through the rapid charging or discharging of programmable resistors. Specifically, for parameters $J_{ik}$, access to their unmodified values is crucial for computing the adjoint nodes. To achieve this, we incorporate additional capacitors (highlighted as red "=" in Figure 3): one dedicated to receiving feedback signals for parameter updates and another to preserve the original values required for adjoint and hidden node calculations. This dual-capacitor configuration ensures accurate gradient computation while enabling efficient, high-speed on-device learning, reinforcing the system's capability for real-time adaptation. The hardware implementation of the newly-introduced adjoint nodes $a_i$ is encoded as a node unit, as depicted in the "$a_i$" section of Figure 3. With EC-Train, the system performs continuous updates within each natural annealing cycle, rapidly reshaping the energy landscape to achieve instant training with exceptional efficiency compared to traditional training on digital processors. The EC-Train training process is detailed as follows:

1. *Initialization:* The capacitor voltages representing inputs and outputs are set to their ground truth values, while the trainable parameters are randomly initialized.

2. *Natural Annealing:* The system undergoes a natural annealing and generates the electrical current $I_m^{in} - I_m^R$, which serves as the feedback signal to adjust the system parameters.

3. *Parameter Adjustment:* The trainable parameters are updated based on the feedback signal.

4. *Iterative Training:* The update of trainable parameters results in a new electrical current $I_m^{in}$ to the node units $y_m$, updating the feedback signal $I_m^{in} - I_m^R$, and instantaneously initiating a new training iteration. This iterative process continues across the training set until convergence is reached.

## 4. Evaluation

As a pioneering effort demonstrating the significant potential of physically embodied dynamical systems, we first evaluate the performance of EADS in graph learning tasks that it is originally designed for, showing the performance of EADS in learning complex functions in real-word problems. Then, we show its potential on other tasks, including PDE solving in scientific computing and approximating important kernels in Large Language Models (LLMs).

**Experimental Platforms.** We conduct our experiments using an NVIDIA A100 40GB SXM GPU for non-dynamical system based baselines, measuring total training time, inference latency per sample, and accuracy. For dynamical system based approaches, we build upon the original hardware embodiment BRIM (Afoakwa et al., 2021), using a custom CUDA-accelerated Finite Element Analysis (FEA) simulator to assess the training time, inference latency, and accuracy. Since the dynamical system based baseline NP-GL (Wu et al., 2024) only achieves inference on dynamical systems, its training time is still measured on an A100 GPU using its own offline training method.

### 4.1. Graph Learning

**Datasets and Baselines.** For complex function learning in real-world problems, we evaluate the performance of EADS in spatial-temporal prediction tasks including six real-world datasets from four applications. (1) Traffic flow prediction with two datasets PEMS04 and PEM08 (Chen et al., 2001). (2) Air quality prediction including PM2.5 and PM10 (Kong et al., 2021). (3) Taxi demand prediction (NYC Taxi): predicting the hourly number of taxi trips (New York City Taxi and Limousine Commission, 2024). (4) Pandemic progression prediction (Texas COVID): predicting the daily number of new cases (Centers for Disease Control and Prevention, 2024). We compare EADS with SOTA spatial-temporal prediction baselines, including Graph WaveNet (Wu et al., 2019), MTGNN (Wu et al., 2020), DDGCRN (Weng et al., 2023), MegaCRN (Jiang et al., 2023), and the dynamical system based method NP-GL (Wu et al., 2024). The number of hidden nodes in EADS is set to 128, and baselines are implemented following the experimental setups detailed in their respective original papers.

**Experimental Results.** We report the test MAE of baselines and EADS in Table 1, where lower values indicate better performance. The results show that EADS outperforms all baselines across all datasets, achieving an average MAE reduction of 23.91%. Notably, EADS reduces MAE by up to 16.29% compared to the best baseline on the Texas Covid dataset. Furthermore, when compared to the dynamical system based baseline NP-GL, EADS achieves an average MAE reduction of 8.08% across all datasets, highlighting the improved system expressivity.

Additionally, Figure 4 presents the training time and inference latency of EADS compared to the baselines, where EADS exhibits remarkable computational efficiency. Specif-

*Table 1.* Spatial-temporal prediction performance in MAE. EADS consistently outperforms all baselines.

| Dataset | PEMS04 | PEMS08 | PM2.5 | PM10 | NYC Taxi | Texas Covid |
|---|---|---|---|---|---|---|
| Graph WaveNet | 20.84 | 15.77 | 1.82 | 1.95 | 10.22 | 82.96 |
| MTGNN | 19.96 | 15.15 | 1.83 | 1.99 | 7.08 | 84.17 |
| DDGCRN | 18.97 | 14.64 | 1.71 | 1.88 | 3.06 | 23.94 |
| MegaCRN | 17.65 | 13.70 | 1.65 | 1.74 | 6.08 | 83.73 |
| NP-GL | 17.07 | 13.51 | 1.62 | 1.73 | 3.03 | 22.04 |
| EADS | 16.92 | 13.43 | 1.53 | 1.62 | 2.46 | 18.45 |

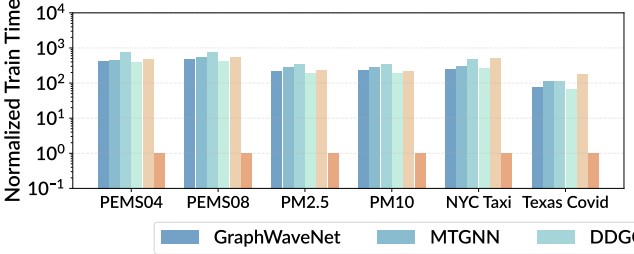 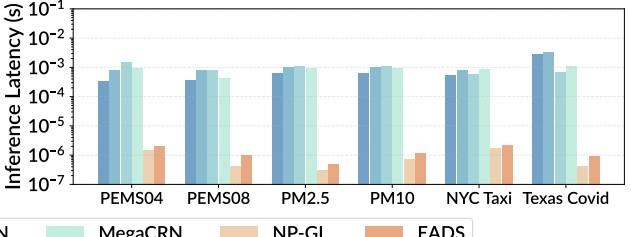

*Figure 4.* Training time and inference latency for spatial-temporal prediction. EADS shows higher efficiency than GPU-based baselines.

ically, EADS delivers an average training speedup of $356\times$ compared to NP-GL. Across all baselines, EADS achieves an average training speedup of approximately $300\times$. Furthermore, in terms of inference latency, EADS delivers an average speedup of around $1000\times$ compared to the baselines executed on GPUs. These substantial improvements in both training and inference efficiency highlight the significant potential of EADS for real-time applications.

### 4.2. PDE Solving

**Datasets and Baselines.** Following (Li et al., 2020a), we evaluate our method on the Burgers' equation and the Darcy Flow equation. The datasets are generated following the same procedure as in (Li et al., 2020a), ensuring consistency in benchmarking. The data is collected on grids of varying spatial resolutions: $16\times16$, $32\times32$, and $64\times64$. The model is trained to learn the mapping from the initial condition (or coefficient field) to the corresponding solution at a specific time, under the same spatial resolution. We compare our method against several established benchmarks, including: NN (Li et al., 2020a), FCN (Zhu & Zabaras, 2018), GNO (Li et al., 2020b), FNO (Li et al., 2020a), and the dynamical system based method NP-GL (Wu et al., 2024). All baselines are implemented following the setups detailed in their respective original papers.

**Experimental Results.** The performance of EADS and the baseline methods on the selected PDEs is summarized in Table 2. Across all evaluated resolutions and PDEs, EADS consistently achieves a lower test MAE than NN, FCN, GNO, and NP-GL. Notably, EADS also exhibits marginally

better accuracy than FNO, underscoring its effectiveness even against advanced operator learning methods.

Beyond accuracy, we compare the training times and inference latencies of all models across the datasets, as illustrated in Figure 5. EADS exhibits exceptional training and inference efficiency, substantially outperforming methods implemented on GPUs. On average, EADS achieves training and inference speedups of around $1000\times$, highlighting its extraordinary computational performance. The results underscore the potential of EADS to accelerate simulations in scientific computing and to support applications demanding

*Table 2.* Test MAE for PDE solving. EADS achieves superior accuracy compared to baselines.

| Methods | Burgers | | |
|---|---|---|---|
| | S1=256 | S2=1024 | S3=4096 |
| NN | $1.26 \times 10^{-3}$ | $1.32 \times 10^{-3}$ | $1.69 \times 10^{-3}$ |
| FCN | $1.37 \times 10^{-4}$ | $1.35 \times 10^{-4}$ | $1.76 \times 10^{-4}$ |
| GNO | $6.95 \times 10^{-5}$ | $6.97 \times 10^{-5}$ | $7.42 \times 10^{-5}$ |
| FNO | $1.41 \times 10^{-5}$ | $1.42 \times 10^{-5}$ | $1.53 \times 10^{-5}$ |
| NP-GL | $1.59 \times 10^{-4}$ | $1.57 \times 10^{-4}$ | $1.83 \times 10^{-4}$ |
| EADS | $1.32 \times 10^{-5}$ | $1.46 \times 10^{-5}$ | $1.51 \times 10^{-5}$ |

| Methods | Darcy Flow | | |
|---|---|---|---|
| | S1=16×16 | S2=32×32 | S3=64×64 |
| NN | $8.48 \times 10^{-6}$ | $8.30 \times 10^{-6}$ | $3.86 \times 10^{-5}$ |
| FCN | $5.83 \times 10^{-6}$ | $6.14 \times 10^{-6}$ | $3.77 \times 10^{-5}$ |
| GNO | $5.71 \times 10^{-6}$ | $5.62 \times 10^{-6}$ | $1.26 \times 10^{-5}$ |
| FNO | $4.72 \times 10^{-6}$ | $3.36 \times 10^{-6}$ | $1.03 \times 10^{-5}$ |
| NP-GL | $6.51 \times 10^{-6}$ | $6.72 \times 10^{-6}$ | $3.65 \times 10^{-5}$ |
| EADS | $4.31 \times 10^{-6}$ | $3.17 \times 10^{-6}$ | $1.12 \times 10^{-5}$ |

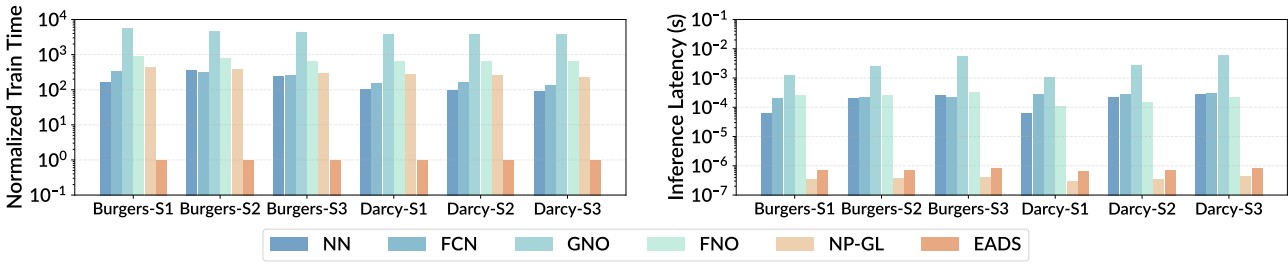

*Figure 5.* Training time and inference latency for PDE solving. EADS is markedly more efficient than GPU-based baselines.

rapid PDE solving, such as interactive design optimization and time-critical decision support systems.

### 4.3. LLMs

**Experimental Setups.** To evaluate the capability of EADS in learning complex functions embedded within advanced ML models, we conducted experiments in the context of LLMs. Specifically, we adopt the GPT-2 small model (Wolf, 2019), which consists of 12 transformer decoders, each containing a causal self-attention kernel. For each attention kernel, we construct a training dataset by extracting input-output pairs from GPT-2's forward pass on the LAMBADA dataset (Paperno et al., 2016), thereby capturing the transformation performed by that kernel. We then train a separate EADS for each of the 12 attention kernels to assess EADS's ability to replicate the underlying complex transformations. During evaluation, we replace one attention kernel in GPT-2 with its corresponding trained EADS while keeping all other GPT-2 components unchanged. The performance of the modified GPT-2 is evaluated using test perplexity (PPL) on the LAMBADA dataset, where lower values indicate better performance. For comparison, we also evaluated NP-GL (Wu et al., 2024), the dynamical system baseline, to provide a reference benchmark.

**Experimental Results.** As shown in Table 3, EADS demonstrates remarkable consistency and robustness across all kernel positions, with only minimal performance degradation ranging from 1.54 to 1.95 PPL points compared to the original GPT-2. This stability indicates that EADS successfully captures and reproduces the complex transformations encoded within each kernel. Notably, EADS substantially outperforms NP-GL across all kernel positions, achieving an average perplexity reduction of approximately 1.49 points. Furthermore, as illustrated in Figure 6, EADS achieves a remarkable training speedup of roughly $\sim 800\times$ compared to NP-GL on GPUs. For inference latency, EADS delivers a speedup of approximately $10^2\times$ over the kernel running on GPUs, underscoring its potential as a promising approach to enhancing LLM efficiency.

## 5. Conclusion

Modern ML methods have demonstrated exceptional capability in approximating various functions, yet their increasing complexity and substantial computational costs pose significant challenges to efficient development. In contrast, nature effortlessly models complex functions through dynamical systems. Inspired by this, we introduce EADS, a nature-inspired ML paradigm that leverages an expressive and self-adaptive dynamical system to learn various functions with unprecedented efficiency. The proposed EADS incorporates hierarchical architectures and heterogeneous dynamics to enhance the expressivity of existing dynamical systems. In addition, we introduce an on-device training method that enables the dynamical system to optimize its parameters using intrinsic electrical signals, thereby achieving exceptional training efficiency. Experiments across functions from diverse domains show that EADS achieves higher accuracy than existing methods while delivering orders-of-magnitude speedups over traditional GPU-based approaches for both inference and training. These results underscore the potential of EADS to overcome the computational bottlenecks across various critical fields.

*Table 3.* Test perplexity (PPL) on LAMBADA.

| Kernels | 1 | 2 | 3 | 4 | 5 | 6 |
|---|---|---|---|---|---|---|
| GPT2 | 35.13 | 35.13 | 35.13 | 35.13 | 35.13 | 35.13 |
| NP-GL | 37.79 | 37.90 | 38.17 | 38.33 | 38.41 | 38.48 |
| EADS | 36.67 | 36.77 | 36.86 | 36.83 | 36.95 | 36.94 |
| Kernels | 7 | 8 | 9 | 10 | 11 | 12 |
| GPT2 | 35.13 | 35.13 | 35.13 | 35.13 | 35.13 | 35.13 |
| NP-GL | 38.41 | 38.38 | 38.72 | 38.31 | 38.79 | 38.77 |
| EADS | 36.75 | 36.93 | 36.96 | 36.87 | 36.94 | 37.08 |

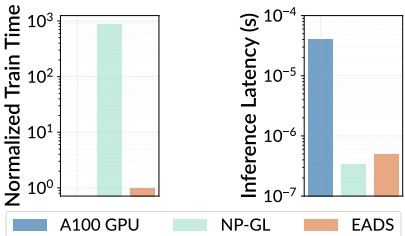

*Figure 6.* Training time and inference latency on LAMBADA.

## Acknowledgements

This work is supported by the U.S. Department of Energy, Office of Science, Office of Advanced Scientific Computing Research, in support of the MEERCAT Microelectronics Science Research Center, under Contract DE-AC05-76RL01830. This work is also supported by DARPA under Contract W912CG25CA007, by NSF under Award No. 2326494, and by NERSC through the perlmutter supercomputer for computational resources.

## Impact Statement

This paper presents work whose goal is to advance the field of dynamical systems for machine learning. There are many potential societal consequences of our work, none which we feel must be specifically highlighted here.

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

# A. Appendix

Below, we present the detailed mathematical derivations of the proposed EC-Train method. The proposed EC-Train approach aims to minimize the difference between $I_m^{in} - I_m^R$ after clamping the output nodes to their ground truth values. In this way, the EC-Train loss function can be formulated as:

$$L = \frac{1}{M} \sum_{i=1}^{M} (I_m^{in} - I_m^R)^2. \tag{13}$$

The error signal $\delta_m$ emerges intrinsically from the dynamical system, serving as natural feedback signals for parameter optimization:

$$\delta_m = \frac{2}{M} (I_m^{in} - I_m^R). \tag{14}$$

Specifically, the gradients with respect to $Q_{mi}$ are then computed as:

$$\frac{\partial L}{\partial Q_{mi}} = \delta_m \cdot h_i. \tag{15}$$

Since output nodes $y_m$ are clamped to their ground truth values during training, $r_m$ essentially acts as a scaling factor for the ground truth signal. To achieve an efficient hardware implementation, we make $r_m = 1$, thereby streamlining both the training and inference processes.

For the inter-hidden-node coupling weight $J_{ik}$ that encoded in the following dynamical process:

$$\frac{dh_i}{dt} = \sigma \left( \sum_{k=1}^{H} J_{ik} h_k \right), \tag{16}$$

where $J_{ik}$ represents inter-node interaction weight and $\sigma(x)$ is a nonlinear function defined as:

$$\sigma(x) = \begin{cases} x, & x > 0, \\ 0, & x \leq 0. \end{cases} \tag{17}$$

Since their final states $(h_1(T), h_2(T), \ldots, h_H(T))$ determines the output node values, we have:

$$\frac{\partial L}{\partial h_i(T)} = \sum_m Q_{mi} \delta_m. \tag{18}$$

To compute parameter gradients, following the adjoint sensitivity method, we introduce the adjoint node $a_i$, satisfying:

$$\frac{da_i}{dt} = -\sum_{l=1}^{H} a_l \frac{\partial f_l}{\partial h_i}, \tag{19}$$

where

$$f_l = \sigma \left( \sum_{i=1}^{H} J_{li} h_i \right). \tag{20}$$

By taking the partial derivative, we obtain:

$$\frac{\partial f_l}{\partial h_i} = \sigma' \left( \sum_{i=1}^{H} J_{li} h_i \right) J_{li}, \tag{21}$$

where $\sigma'(x)$ is the derivative of $\sigma(x)$, given by:

$$\sigma'(x) = \begin{cases} 1, & x > 0, \\ 0, & x \leq 0. \end{cases} \tag{22}$$

Consequently, the adjoint node evolves as:

$$\frac{da_i}{dt} = -\sum_{l=1}^{H} a_l \sigma' \left( \sum_{i=1}^{H} J_{li} h_i \right) J_{li} = -\sum_{l=1}^{H} a_l I_l J_{li}. \tag{23}$$

with the initial value:

$$a_i(T) = \frac{\partial L}{\partial h_i(T)} = \sum_m Q_{mi} \delta_m. \tag{24}$$

Here, $I_l$ is a boolean indicator defined as:

$$I_l = \sigma' \left( \sum_{s=1}^{H} J_{ls} h_s > 0 \right) = \mathbb{I} \left( \sum_{s=1}^{H} J_{ls} h_s > 0 \right). \tag{25}$$

Using the adjoint nodes, the gradient of $L$ with respect to $J_{ik}$ is computed as:

$$\frac{\partial L}{\partial J_{ik}} = -\int_T^0 a_i \frac{\partial f_i}{\partial J_{ik}} = -\int_T^0 a_i \sigma' \left( \sum_{k=1}^{H} J_{ik} h_k \right) h_k dt = -\int_T^0 a_i I_i h_k dt. \tag{26}$$

For the input-to-hidden dynamical process:

$$\frac{dh_i}{dt} = \sum_{j=1}^{N} P_{ij} x_j - r_i h_i, \tag{27}$$

where $h_i$ is the dynamical node, $x_j$ is the fixed input, $P_{ij}$ is the parameter. According to Kirchhoff's current law, $h_i$ reaches a steady state when its incoming current $I_i^{in} = \sum_{j=1}^{N} P_{ij} x_j$ is balanced by the current through its internal resistor, $I_i^R = r_i h_i$. Given that each input $x_j$ is clamped to its ground-truth value, the equilibrium condition implies a stable hidden state value of

$$h_i = \sum_{j=1}^{N} P_{ij} x_j, \tag{28}$$

assuming $r_i = 1$ without loss of generality, as it functions as a scaling factor. These equilibrium states then serve as the initial states for the inter-hidden dynamical process in Eq. 16, thus the error signal propagated back from the inter-hidden stage to this input-to-hidden stage is $\delta_i = a_i|_{t=0}$. Consequently, the gradient with respect to the projection weight $P_{ij}$ is given by:

$$\frac{\partial L}{\partial P_{ij}} = \delta_i \cdot x_j. \tag{29}$$

