# OpenReview forum: "An Expressive and Self-Adaptive Dynamical System for Efficient Function Learning"
_ICML.cc/2025/Conference — ICML 2025 poster_

### Official Review · Reviewer_z4kd · 2025-03-07

**Overall Recommendation:** 2

**Summary:**

This paper introduces EADS, an framework for learning equations efficiently
. EADS is inspired by the efficiency of natural systems in learning and solving complex equations. The authors argue that EADS overcomes the limitations of traditional ML methods, such as high computational cost and complex models. They demonstrate EADS's superior performance in various tasks, including graph learning, PDE solving, and LLMs.

**Claims And Evidence:**

The main claim is that EADS achieves higher accuracy than traditional ML methods. This is justified by experimental results on graph learning, PDE solving, and LLMs show that EADS outperforms baselines in terms of accuracy. The proposed method also achieves much higher efficiency compared with those methods running on dedicated gpus, for both inference and training.

**Essential References Not Discussed:**

n/a

**Experimental Designs Or Analyses:**

In figure 6, EADS achieves much faster training time than NG-PL due to training implementation on device. In section 4.3, it says "we replace a single decoder block with its corresponding EADS model while keeping all other components of GPT-2 unchanged", is GPT-2 also implemented on the device? can you specify the details?

There also lacks details regarding the architecture of the proposed method for each experiment setting (e.g., number of parameters, hidden units, input/output dimensions.)

**Methods And Evaluation Criteria:**

The performance of EADS is evaluated based on accuracy, training time, inference latency, and energy consumption.
I have concerns regarding the evaluation baselines.

1.Regarding accuracy (e.g, table 2), it is difficult to judge the effectiveness of the proposed method without mentioning the number of parameters in each method.

2.Regarding energy consumption (e.g section 4.4), why a dedicated all purpose gpu is introduced for comparison with the proposed hardware customized to the specific model design? A fair comparison should include NP-GL, which also significantly reduce the energy cost on CMOS-compatible computers.

3.The above concern can also apply to training time, as the major contribution is from not using gpu instead of the proposed method.

4. Including theoretical computational complexity could be a more suitable evaluation criterion for all algorithms, as it is independent of hardware constraints.

**Other Comments Or Suggestions:**

n/a

**Other Strengths And Weaknesses:**

n/a

**Questions For Authors:**

How does EADS compare to other methods in terms of scalability and flexibility? Do we need to customize the hardware if the model is modified?

Why the method is called "dynamical system" rather than "dynamical system model" or more specific models like Ising model? EADS is essentially a model to capture the system behavior, not the system itself.

I'm curious under what circumstance on-devide training is needed?

**Relation To Broader Scientific Literature:**

n/a

**Theoretical Claims:**

n/a

---

> ### Author Rebuttal · Authors · 2025-04-01
>
> We sincerely appreciate your insightful comments. We will address your questions below.
>
> **1.The number of parameters**
> We will report the number of parameters in the Appendix. Due to the character limit, we present parameters for a dataset below. For the PEMS04 dataset: Graph WaveNet: 250,689; MTGNN: 555,169; DDGCRN: 567,109; MegaCRN: 316,225; NP-GL: 188,498; EADS: 94,976.
>
> **2.Include NP-GL in Comparison**
>
> NP-GL supports inference on a dynamical system, its training is conducted offline on GPUs. According to [1], NP-GL operates with similar power for inference to EADS. Both methods achieve similar inference latency, resulting in comparable inference energy efficiency. However, for training, compared to NP-GL, EADS achieves an average ~$10^{2}$ training speedup, ~ $10^5$ greater energy efficiency.
>
> **3.Include Theoretical Computational Complexity**
> The computing paradigm of EADS is inherently different from that of digital methods, rendering their complexities not directly comparable. Specifically, GPU-based methods execute explicit instructions sequentially, while EADS operates via natural annealing—where electrons (dis)charge capacitors to seek equilibrium. To provide a clear understanding, we define one operation in our DS as a single substantial (dis)charge event of a capacitor, taking ~0.2 ns. For a digital processor, a single instruction execution requires ~0.25 ns. Given that the operation execution times are comparable, we evaluate computational complexity by comparing their total number of operations. Our results show that EADS requires ~$10^3$ operations per inference, while GPU-based solutions require ~$10^5$ operations per inference. We will incorporate detailed discussion.
>
> **4.Experiment Details on LLMs**
> We design our LLM experiments to assess EADS’s ability to learn the complex transformations embedded within each decoder of GPT-2. The entire GPT-2 model is not implemented on EADS; rather, only one decoder is implemented by EADS. Although our system has the potential to implement all decoders, such an extension is beyond the current scope of our work. We will provide details in the manuscript.
>
> **5.Include the Settings of EADS**
> Thanks for your valuable suggestion. We will add a comprehensive section in the Appendix specifying EADS configurations.
>
> **6.Scalability and Flexibility of EADS**
> - Scalability: Dynamical systems have demonstrated strong scalability through single-chip and multi-chip solutions, as evidenced in [2-3]. Specifically, [2] propose a single-chip solution, while [3] explored multi-chip solutions. Overall, the scalability of dynamical system machines is well-founded.
> - Flexibility of EADS: Since EADS enables on-device training, its parameters can be dynamically adjusted to suit different problems without hardware modifications, making it highly versatile.
>
> **7.Why Is the Method Called Dynamical System**
> We refer to EADS as a “dynamical system” because EADS is not merely a model—it is a software-hardware co-design implemented using programmable electronic components.
>
> **8.Under What Circumstances Is On-Device Training Needed?**
> On-device training is essential for several reasons:
> 1. Enabling a New Computing Paradigm: Instant on-device training extends the exceptional computational power of dynamical systems beyond inference to include training, thereby establishing a new AI paradigm with remarkable efficiency.
> 2. Applications Requiring Real-Time Adaptation or Facing Training Costs:
> - By implementing training directly on-device, EADS can update parameters automatically in response to new data in real time without the delays associated with off-device training, ensuring accurate predictions as underlying patterns evolve.
> - In scientific computing, the potential of data-driven equation solving is often hindered by prohibitive computational costs. As ML models scale to achieve higher accuracy, training resources may exceed those required by traditional numerical solvers, limiting their practical benefits. Our on-device training directly addresses this challenge.
> 3. Advancing AI Paradigm Development: Recent research suggests that collocating inference and training on the same hardware can significantly reduce training costs while more closely mirroring biological intelligence [4]. There is also evidence indicating that the human hippocampus functions as a dynamical system with collocated training and inference [5]. Our proposed on-device training method offers a meaningful exploration in this domain.
>
> [1] Extending power of nature from binary to real-valued graph learning in real world
>
> [2] DS-GL: Advancing Graph Learning via Harnessing Nature’s Power within Scalable Dynamical Systems
>
> [3] Increasing ising machine capacity with multi-chip architectures
>
> [4] The forward-forward algorithm: Some preliminary investigations
>
> [5] Attractor dynamics in the hippocampal representation of the local environment

---

> > ### Comment · Reviewer_z4kd · 2025-04-04
> >
> > I thank the authors for the rebuttal work.
> >
> > As the proposed efficiency improvement arises from two sides, the numerical algorithm and hardware efficiency, my major concern still lies on the issue that the paper lacks a clear disentanglement of the two contributions, not to say the experiments are eventually evaluated from the GPU.
> >
> > After checking other reviews, unfortunately it looks like all the reviewers (including me) are familiar with the algorithm side (hamiltonian, equation discovery), not the hardware side. I will keep my score now and am looking forward to discussing this further with other reviewers and AC.
> >
> > PS: from your rebuttal, why does EAD only take 10^3 operations per inference but with around 94k parameters? Are they not activated?

---

> > > ### Author Response · Authors · 2025-04-07
> > >
> > > Thank you very much for your insightful comment! We fully understand the concerns and have provided structured clarifications below.
> > >
> > > **Why hardware/software in traditional codesign can be disentangled?**
> > >
> > > Traditional AI development follows a top-down paradigm, where the AI community designs universal algorithms that can be deployed across a wide range of digital processors, such as GPUs/TPUs/FPGAs. In this paradigm, the algorithm remains useful as hardware changes—highlighting a decoupled relationship between algorithm and hardware. This approach, where the AI algorithm lives/exists independently of the hardware that runs it, has been referred to as “immortal computing” [1] by Prof. Geoffrey Hinton.
> > >
> > > While this separation offers versatility, it also inevitably results in mismatches between AI algorithmic demands and hardware capabilities, further leading to inefficiencies and high computing costs in AI computing. To mitigate this, the computer systems community has focused on accelerating AI by narrowing this gap through algorithm-hardware co-design. However, in most existing approaches, the co-designed algorithm is still based on “immortal computing”, conceived as a sequence of instructions (e.g., MAC operations), hence can be executed by most digital hardware. This nature makes it relatively straightforward to disentangle contributions: algorithmic complexity is measured by the number of instructions, while hardware performance is measured by instruction throughput.
> > >
> > > **Why is entangled hardware/software important?**
> > >
> > > Traditional AI algorithm-hardware co-design remains vital to efficient AI development. However, it is important to note that we are entering an exciting yet challenging era of AI. As Moore’s Law approaches its limit—constraining further gains in computational power—while AI workloads continue to grow exponentially, exploring fundamentally more efficient computing paradigms is crucial for the long-term sustainability of AI development.
> > >
> > > One promising direction is the “mortal computing” paradigm, as introduced by Prof. Geoffrey Hinton (Sec. 9 in [1]), where hardware and software are tightly entangled with minimal mismatch—much like biological intelligence systems such as the brain. Evidence from prestigious scientific articles [2] shows that biological intelligence (brain):
> > >
> > > 1. Computes through stochastic and continuous processes, rather than deterministic, instruction-based sequences (e.g., A × B + C × D), and
> > >
> > > 2. Is inherently mortal—i.e., the "model" and the organ that realizes it are inseparable. For example, a monkey’s intelligence cannot be ported to a human brain, and individual differences in brain structures result in variations in human intelligence.
> > >
> > > **Why is our work inherently hard to disentangle?**
> > >
> > > Our work embraces the mortal computing paradigm. The hardware is a physical dynamical system that naturally evolves toward equilibrium (minimum energy) through a stochastic and continuous process known as natural annealing. The algorithm is grounded in the natural annealing process, rather than relying on a predefined sequence of instructions.
> > >
> > > This deep coupling between algorithm and hardware is key to the exceptional efficiency of our approach—but it also makes it difficult to isolate the contributions of hardware and software. In contrast to GPUs, which perform inference for universal models (e.g., neural networks) through step-by-step execution of instruction sequences, our system performs inference (EADS) via natural annealing—a process in which electrical current and voltage evolve continuously, driven by capacitor (dis)charge events, until equilibrium is reached.
> > >
> > > **Computational complexity and 94k parameters?**
> > >
> > > For a fair comparison between EADS and traditional neural networks, accuracy and total execution time are the most meaningful metrics. However, to provide an intuitive sense of computational complexity, we identify the primitive/basic computing unit in our system as a substantial capacitor (dis)charge event, which drives the evolution of the system toward equilibrium. Each such event takes approximately 0.2 nanoseconds—comparable to the execution time of a single instruction on a 4 GHz processor (~0.25 ns).
> > >
> > > Thus, we propose defining a single substantial (dis)charge event as one operation in our system. However, it's important to emphasize that, unlike in digital processors where each instruction typically involves only one or a few parameters, each operation in our system engages all 94K parameters simultaneously. In every operation, these parameters collectively influence the charging and discharging of capacitors, jointly driving the evolution of electrical voltage and current, and ultimately pushing the system toward equilibrium.
> > >
> > >
> > > [1] Hinton, Geoffrey. The forward-forward algorithm: Some preliminary investigations. arXiv.
> > >
> > > [2] Wills, Tom J., et al. Attractor dynamics in the hippocampal representation of the local environment. Science 308.5723.

---

### Official Review · Reviewer_17Sp · 2025-03-08

**Overall Recommendation:** 3

**Summary:**

The paper proposes an Expressive and self-Adaptive Dynamical System (EADS) that can learn a wide range of equations with efficiency. The authors propose an efficient on-device learning method that leverages intrinsic electrical signals to update parameters, making EADS self-adaptive at a reduced cost. The authors explore the accuracy of EADS and compare it to existing works, showing that EADS can provide speedups and energy efficiency over other methods.

## update after rebuttal

In view of the authors rebuttal, I updated my score 2 -> 3.

**Claims And Evidence:**

The paper show evidence of good performance of the proposed method.

**Essential References Not Discussed:**

The paper does not discuss any reference in the field of equation discovery, also known as symbolic regression. The authors should discuss how their method relates to existing methods in the field of equation discovery.

**Experimental Designs Or Analyses:**

I did not run experiments to verify the results.

**Methods And Evaluation Criteria:**

The method is evaluated on a set of benchmark datasets. Some baselines are missed in the experimental evaluation. Notably, a genetic programming method should be included in the comparison and the Finite Element Method (FEM) for the PDE solving task.

**Other Comments Or Suggestions:**

1. The paper repeatedly uses the idea of a dynamical system being able to "learn" equations. (e.g., "... dynamical systems effortlessly learn complex equations ...", "Natural systems effortlessly learn and solve complex equations through inherent dynamical processes.", etc). In mathematical physics, the term "dynamical system" refers to a system that evolves over time according to a set of rules, typically described by a system of differential equations. (From Wikipedia: "In mathematics, a dynamical system is a system in which a function describes the time dependence of a point in an ambient space, such as in a parametric curve.") Therefore, the equation that describes the dynamics of a dynamical system is not "learned" by the system, but rather is a fundamental property of the system. It is very important that the authors clarify this use of the term "learn" in the context of dynamical systems.

2. Rewrite the line 327: "...we first evaluate the performance of EADS in graph learning tasks that it is originally deployed, showing the performance of EADS ...". Do you mean "in graph learning tasks that it is originally designed for"?

3. A definition of "inference latency" will be pertinent for the reader to understand the results presented in the paper. The authors should provide a definition of this term in the paper.

4. Line 306 "For hard-to-define equation learning in real-world problems, ..." What do the authors mean by "hard-to-define equation learning"? This sentence is vague and should be clarified.

**Other Strengths And Weaknesses:**

- Strengths
    - The paper addresses the important question of efficiency in inference and training in machine learning.
- Weakness
    - In my opinion, the language in the paper is not precise and too colorful for a scientific paper. For example, "Nature presents an elegant solution to this computational crisis." or "Can we harness dynamical systems to create a nature-powered ML paradigm that learns and solves equations with revolutionary efficiency?". I would suggest the authors to use a more formal language.

**Questions For Authors:**

1. (Abstract) "While modern machine learning (ML) methods are powerful equation learners, their escalating complexity and high operational costs hinder sustainable development." What do the authors mean by "equation learners"? This is not a standard term in the ML literature. Is a LLM a "equation learner"? What do the authors mean by "sustainable development" in this context? This sentence is vague and should be clarified.

2. (Section 2.1). " Given a well-trained Hamiltonian that accurately captures the correlation between inputs and outputs, ..." How is the Hamiltonian trained? Can the authors provide an example on how to do this for a toy problem? I think that will help the reader to understand the concept better.

3. (Section 4.2) The authors present examples of PDE Solving. How does the Hamiltonian of the system is trained in this case? Can the authors provide an example on how to do this for a simple Parabolic PDE?

4. Experimental Results. A critical aspect of learning equations is the ability to inspect the learned equations and assess their interpretability. Can the authors provide some examples of the learned equations and discuss their interpretability?

5. Table 1. How does the method compare to other methods in the field of equation discovery (aka symbolic regression)? In particular, a well-stablihed method in the field is genetic programming. How does the proposed method compare to genetic programming in terms of accuracy and efficiency?

6. (4.2 PDE Solving) The authors should compare with a Finite Element Method (FEM) solver for the PDEs. How does the proposed method compare to a FEM solver in terms of accuracy and efficiency?

**Relation To Broader Scientific Literature:**

The paper does not emphasize enough the relation of the proposed method with the literature on equation discovery (aka symbolic regression). The authors should discuss how their method relates to existing methods in the field of equation discovery.

**Theoretical Claims:**

Theoretical claims are not discussed in the paper.

---

> ### Author Rebuttal · Authors · 2025-04-01
>
> We appreciate your thorough review and constructive feedback. We address each concern in detail.
>
> **1.Relation to Equation Discovery**
> We clarify that our work fundamentally differs from equation discovery: rather than discovering equations, our work focuses on efficiently solving equations with high speed and low computational cost while maintaining high accuracy. EADS works by optimizing its parameters through the proposed on-device training method to capture the joint distributions between inputs and outputs. Once trained, it can efficiently generate solutions for new inputs. In PDE solving, equation discovery methods would attempt to identify the underlying equations, while EADS aims to generate solutions to PDEs under varying inputs (different coefficients or initial conditions) as described in [1]. Importantly, our work extends to solve analytically intractable equation. We will add a subsection in the Related Work to state the relation.
>
> **2.Clarification of Learn in Dynamical Systems**
> We apologize for any confusion. As noted in Q1, learning is indeed misleading since our focus is on solving equations efficiently rather than learning them. We agree that EADS do not learn their inherent dynamics. However, it's worth noting that natural dynamical systems do adjust the parameters within their dynamics. A typical example is the human hippocampus, which functions as a dynamical system and adjusts synaptic strengths (i.e., parameters) between neurons [2]. We will revise the manuscript to replace “learn equations” with “tune the parameters within dynamics.”
>
> **3.Formality of language, Sentence Clarifications and Key Term Definitions**
> We will revise the manuscript to use more precise language.
> - Line 327: Revised to “in graph learning tasks that it is originally designed for.”
> - Inference Latency: Defined as the time delay between submitting an input to a trained model and receiving its output.
> - Hard-to-Define Equation Learning: Revised to “For complex real-world problems where underlying equations are unknown or not readily expressible in closed form.”
> - Abstract: Revised to “While modern machine learning methods approximate complex functions, their escalating complexity and computational demands pose challenges to efficient deployment.”
>
> **4.Hamiltonian Training and Example**
> Training Hamiltonian involves optimizing its parameters to capture correlations between nodes. Consider $\mathcal{H} = -\sum_{i\neq j}^{N} J_{ij} x_i x_j + \sum_{i=1}^{N} h_i x_i^2$, with trainable parameters $J,h$. Using a conditional likelihood method [3], the estimated state at equilibrium is: $\hat{x_i} = \frac{1}{2h_i}\sum_{j\neq i}^{N} (J_{ij}+J_{ji}) x_j.$ Then, the MSE loss is minimized via standard backpropagation to optimize parameters. Detailed explanation and example will be included in Section 2.1.
>
> **6.EADS Training in PDE Solving**
> In PDE solving, EADS approximates solutions under varying inputs (coefficients or initial conditions). Training is optimizing system parameters ($P,J,Q$) to capture the correlations between inputs and outputs. In one of the Darcy Flow example [1], we solve the 2D Darcy Flow equation on the unit square with Dirichlet conditions. Each training sample comprises a 16×16 grid of coefficient a(x) and its corresponding solution u(x). EADS learns mapping from a to u, then rapidly generate solutions without iterative solvers. A detailed description will be added in the Appendix.
>
> **7.Interpretability and Comparison with Equation Discovery and FEM**
> As explained in our response to Q1, our method does not aim to derive explicit equations from data.
> Since the goal and required training data are significantly different between our method and methods in equation discovery methods, it is difficult to incorporate genetic programming based methods in our evaluation for a fair comparison. To provide a reference, a method in the genetic programming-based equation discovery domain that also evaluated on the Burgers' equation achieves an MSE of $4.33×10^{-5}$ [4], while our method achieves an average MSE of $9.37×10^{-6}$ under our evaluation.
> Regarding the comparison with FEM, our evaluation employs outputs from the FEM as ground truth, so we only compare their efficiency. On average, the inference latency of EADS is at the level of 10e-7 seconds, while the latency of typical FEM generally exceeds 10e-3 seconds. Therefore, the efficiency of EADS is significantly better than FEM, especially on complex PDEs with higher resolutions.
>
> [1] Li, Z., et al. Fourier Neural Operator for Parametric Partial Differential Equations. ICLR.
> [2] Wills, T.J., et al. Attractor dynamics in the hippocampal representation of the local environment. Science, 2005.
> [3] Wu, C., et al. Extending power of nature from binary to real-valued graph learning in real world. ICLR.
> [4] Chen, Y., et al. Symbolic genetic algorithm for discovering open-form partial differential equations (SGA-PDE). Physical Review Research.

---

> > ### Comment · Reviewer_17Sp · 2025-04-02
> >
> > I appreciate the authors' detailed response to my comments.
> >
> > The acknowledgement of the misleading use of the term "learn" in the context of dynamical systems and subsequent revision of the manuscript to clarify this point will improve the clarity of the paper. This is specially important, considering that the paper is entitled "An expressive and self-adaptive dynamical system for efficient **equation learning**".
> >
> > I also appreciate the authors' willingness to revise the manuscript to use more precise language and clarify the sentences I pointed out.
> >
> > The authors' response to the questions regarding the PDE solving task is helpful.
> >
> > I will update my overall recommendation to reflect the authors' clarifications.

---

> > > ### Author Response · Authors · 2025-04-03
> > >
> > > Dear Reviewer 17Sp,
> > >
> > > Thank you for your thoughtful feedback and for taking the time to review our response. We sincerely appreciate your acceptance of our work. Your constructive insights have been invaluable in improving the quality of the manuscript. We will carefully follow your suggestions and improve the manuscript accordingly. Please do not hesitate to let us know if there are any remaining concerns or additional details that we can address to further improve the manuscript.
> > >
> > > Thank you once again for your thorough and valuable review.
> > >
> > > Best regards,
> > >
> > > The Authors

---

### Official Review · Reviewer_yCbr · 2025-03-13

**Overall Recommendation:** 4

**Summary:**

The authors propose a method to learn equations efficiently from data. Current methods such as neural networks have high complexity and high operational cost which hinders widespread applications. Recently electronic dynamic systems have shown great promise in solving simple learning problems with great efficiency. However, current electronic dynamic systems lack sufficient expressivity to learn complex equations, and also lack effective training support. To mitigate the current limitations of electronic dynamic systems for learning, the authors propose Expressive and self Adaptive Dynamic System (EADS) that integrates hierarchical architecture and heterogeneous dynamics to increase expressivity and also propose an on-device learning method for efficiency. Experiments show that EADS has lower training times and higher/comparable accuracy compared to baselines.

**Claims And Evidence:**

The authors claim their method provides low training times and more expressive power. Experiments show that the training times are lower. The accuracy is clearly higher for one experimental setup, while for others they are comparable. The comparable accuracy experiments can use some additional possible explanations.

**Essential References Not Discussed:**

None

**Experimental Designs Or Analyses:**

I looked at the experimental designs, at quick glance the experimental designs look good, however I haven’t checked the details thoroughly.

**Methods And Evaluation Criteria:**

The proposed method makes sense, as the authors clearly identify 2 gaps in current literature, and their proposed method incorporates possible remedies for the identified gaps. The training times are measured for efficiency comparison, while accuracy is measured for expressivity comparison.

**Other Comments Or Suggestions:**

None

**Other Strengths And Weaknesses:**

The authors present a very good introduction, clearly identifying gaps in current literature and justifying their proposed solutions. I have few suggestions that the authors can consider for improving the current manuscript:
1. Equation learning can be confused with symbolic equation learning/symbolic regression, I think a better name could be function learning. To avoid confusion, the authors can add a line or two in the introduction.
2. It will be great if the authors can provide a simplified visual diagram of EADS workflow, that will be very helpful to interested readers.
3. The captions can be more detailed, highlighting the key message i.e., figure shows EADS is more efficient

**Questions For Authors:**

None

**Relation To Broader Scientific Literature:**

Energy efficient training is a major concern at present for complex machine learning models, the authors work can significantly push the limit of training efficiency using hardware optimizations through dynamic systems. As scientific literature in many domains now heavily relies on machine learning methods and model training, this can have a broad impact on overall literature.

**Theoretical Claims:**

None

---

> ### Author Rebuttal · Authors · 2025-04-01
>
> We sincerely appreciate your positive feedback and constructive suggestions. Below, we address each of your points in detail to further improve the manuscript.
>
> **1.Additional explanations for comparable accuracy experiments**
>
> Thank you for this insightful suggestion. In our evaluations, we compare EADS with NP-GL and deliberately selected state-of-the-art digital baselines that have consistently demonstrated exceptional accuracy through rigorous validation in prior research. Although the accuracy differences may appear similar in magnitude, the actual improvements are significant. Specifically, compared to NP-GL, EADS achieves an average MAE reduction of 8.92% on spatial-temporal prediction, a 71.4% MAE reduction in PDE solving, and a 6.32% reduction in PPL on LLM tasks. Furthermore, compared to the best digital baseline, EADS yields a 15.10% MAE reduction on spatial-temporal prediction, and a 4.62% MAE reduction on PDE solving, while substantially reducing computational demands. We will expand our analysis with additional explanations to better highlight the advantages of EADS.
>
> **2.Terminology clarification**
>
> Thank you for highlighting this potential confusion and for your valuable suggestion. We agree that the term “function learning” offers a more precise description of our work. We will change “equation learning” to “function learning” throughout the manuscript and differentiate our focus from symbolic regression in the Introduction.
>
> **3.Add visual diagram for EADS workflow**
>
> We appreciate your valuable suggestion. We will include a new figure in the Methodology section to illustrate the overall EADS workflow. As we cannot directly upload figures in the rebuttal, we provide a detailed description of the proposed diagram below. The diagram will be divided into two main components:
> - **Expressivity Enhancement:** We will visualize the hierarchical structure and heterogeneous dynamics with clear illustrations of information flow. This visual will demonstrate how EADS progressively refines the input through multiple processing stages with heterogeneous dynamics.
> - **Instant On-Device Training:** This part will detail the parameter adjustment process. It will depict how intrinsic electrical signals are used as feedback to drive learning, illustrating the feedback loop where the output nodes’ electrical currents guide rapid, on-device parameter updates.
>
> **4.Enhanced figure and table captions to highlight key messages**
>
> Thank you for recommending more informative captions. In our revision, we will update the captions to explicitly highlight the key performance takeaways:
> - **Table 1:** Spatial-temporal prediction performance in MAE. EADS consistently outperforms all baselines across all datasets (best results in bold).
> - **Figure 4:** Training time and inference latency for spatial-temporal prediction. EADS demonstrates significantly higher efficiency than GPU-based baselines.
> - **Table 2:** Test MAE for PDE solving. EADS achieves superior accuracy compared to all baselines, with marginal improvements over FNO.
> - **Figure 5:** Training time and inference latency for PDE solving. EADS is markedly more efficient than GPU-based baselines.
> - **Table 3:** Test perplexity (PPL) on LAMBADA. EADS significantly outperforms NP-GL.
> - **Figure 6:** Training time and inference latency on LAMBADA. EADS exhibits superior efficiency compared to GPU-based methods.

---

### Decision · Program_Chairs · 2025-05-01

**Decision:**

Accept (poster)

**Comment:**

The proposed method improves on the high complexity of existing methods for equation learning. The proposes an interesting novelty in combining algorithmic and hardware aspects to equation learning. The empirical evaluation demonstrated the improved efficiency of the method.